# Low vs standardized dose anticoagulation regimens for extracorporeal membrane oxygenation: A meta-analysis

**Xiaochai Lv**[1,2], **Manjun Deng**[3], **Lei Wang**[1,2], **Yi Dong**[1,2], **Liangwan Chen**[1,2]*, **Xiaofu Dai**[1,2]

**1** Department of Cardiac Surgery, Fujian Medical University Union Hospital, Fujian, China, **2** Department of Fujian Key Laboratory of Cardio-Thoracic Surgery, Fujian Medical University, Fujian, China, **3** Department of Hepatopancreatobiliary Surgery, Mengchao Hepatobiliary Hospital of Fujian Medical University, Fujian, China

* chenliangwan2019@163.com.

**Data Availability Statement:** All relevant data are within the manuscript and its Supporting Information files.

## Abstract

### Background

To compare the safety and efficacy of low-dose anticoagulation (LA) with that of standardized dose anticoagulation (SA) for patients supported with extracorporeal membrane oxygenation (ECMO).

### Methods

PubMed, MEDLINE, the Cochrane Library, and Web of Science were screened for original articles. Screening was performed using predefined search terms to identify cohort studies reporting the comparison of LA with SA in patients supported with ECMO from Nov 1990 to Jun 2020. The effect size was determined by the odds ratio (OR) with the 95% confidence interval (CI).

### Results

An analysis of 7 studies including a total of 553 patients was performed. LA (Low-heparin group) was administered to 255 patients, whereas the other 298 patients received SA (Full-heparin group). The incidence of gastrointestinal tract hemorrhage (OR 0.36, 95% CI 0.20–0.64) and surgical site hemorrhage (OR 0.43, 95% CI 0.20–0.94) were significantly lower in patients who underwent LA compared with that in those who underwent SA. The rates of hospital mortality (OR 0.81, 95% CI 0.42–1.56), successfully weaning off of ECMO (OR 0.80, 95% CI 0.30–2.14), pulmonary embolism (OR 0.79, 95% CI 0.24–2.65), intracardiac thrombus (OR 0.34, 95% CI 0.09–1.30), intracranial hemorrhage (OR 0.62, 95% CI 0.22–1.74), and pulmonary hemorrhage (OR 0.77, 95% CI 0.30–1.93) were similar between the two groups.

### Conclusions

This meta-analysis confirms that LA is a feasible and safe anticoagulation strategy in patients supported by ECMO. Future studies should focus on the long-term benefits of LA compared with SA.

**Funding:** This work was supported by the National Natural Science Foundation of China (No. 81370414, 81670438).

**Competing interests:** The authors have declared that no competing interests exist.

## Introduction

Extracorporeal membrane oxygenation (ECMO) is a rescue therapy method for cardiopulmonary function, which was originally developed by Dr. Bartlett in the early 1970s as an improvement to cardiopulmonary bypass [1]. Therefore, the use of an anticoagulant strategy with ECMO is partly consistent with the concept of cardiopulmonary bypass. Based on expert opinion and consensus, the Extracorporeal Life Support Organization (ELSO) published guidelines in 2014 that recommended the use of heparin for systemic anticoagulation with ECMO support to prevent thrombosis, and the activated clotting time (ACT) target value is 180–220 seconds [2]. Even though anticoagulation has been used to prevent clots in the ECMO cannulae, oxygenator and tubing [3], excessive anticoagulation may cause hemorrhagic complications, which have a significantly higher risk of mortality [4].

Bleeding is a frequent complication of ECMO, which can be catastrophic, including intracerebral hemorrhage, surgical site bleeding, and gastrointestinal hemorrhage [5, 6]. In a series of surveys, Dalton et al. and Mazzeffi et al. evaluated bleeding complications in ECMO and found that it occurred in 27% to 60% of adult patients [7, 8], which portends considerable high mortality. However, it is necessary to note that inadequate anticoagulation may lead to thrombosis. Notably, the advancement of technology, including centrifugal pumps, oxygenators and biocompatible circuits, have theoretically lowered the risk of thromboembolic complications. Consequently, in an attempt to offer less hemorrhagic complications to patients with ECMO, low-dose anticoagulation (LA) strategies have been gradually proposed [9–13].

LA offers advantages such as fewer bleeding complications, decreased blood transfusions, and superior survival when compared with standardized dose anticoagulation (SA) strategies [14–16]. Nevertheless, it is worth noting that some studies reporting a degree of microembolic events may occur in ECMO patients, but the clinical significance of this potential microembolic burden has not yet been determined [17]. Furthermore, Lamarche et al. provided evidence that a low incidence of oxygenator failure (9%) in patients supported on Veno-Arterial (VA) ECMO without anticoagulation [18]. Wood KL et al. have shown encouraging data that the lack of systemic anticoagulation did not increase thrombotic events in the ECMO circuit [16].

Recently, numerous centers have published their experiences with LA; however, the equivalence or benefit between LA and SA is still a topic of debate in terms of patients supported on ECMO [19]. With the aim of determining whether LA is superior to SA in ECMO, we conducted a meta-analysis comparing hemorrhagic complications (gastrointestinal tract hemorrhage, surgical site hemorrhage, intracranial hemorrhage, pulmonary hemorrhage, and ECMO cannula site bleeding), thrombotic complications (deep vein thrombosis, pulmonary embolism, clots in the oxygenator and pump, and intracardiac thrombus), hospital mortality, oxygenator exchange and the successfully weaned off ECMO rates for the patients who required ECMO.

## Methods

### Search strategy and literature selection

A systematic review and meta-analysis was performed in this study. We searched PubMed, MEDLINE, the Cochrane Library, and Web of Science to identify observational studies and RCTs from Nov 1990 to Jun 2020. Studies were restricted to English. Key words used in PubMed were as follows: ("extracorporeal membrane oxygenation" OR "extra-corporeal membrane oxygenation" OR "ECMO") AND ("anticoagulation") AND ("heparin") AND ("low-dose anticoagulation" or "low anticoagulation" or "low-dose heparin" or "sparing anticoagulation") AND ("standard anticoagulation" or "systemic anticoagulation" or "therapeutic

anticoagulation" or "conventional heparin treatment"). Search strategies for other databases were modified based on the requirements of each database. Any potentially eligible studies were screened manually through the references of the included studies, relevant meta-analyses, reviews and guidelines, or contacted authors.

The evaluation of all searched results were based on the Preferred Reporting Items for Systematic Reviews and Meta-Analyses (PRISMA) statement [20]. We selected the original research by the process of viewing titles, abstracts, and full papers. We only included studies that were concerned with the comparison of LA with SA in patients supported by ECMO. Patients were included in the SA (Full-heparin group) if they were started on a continuous infusion of anticoagulant after initiation of ECMO, which was monitored with ACT with a target of 180–220 seconds or activated partial thromboplastin time (aPTT) target of 50–70 seconds. The Low-heparin group was supported with an LA protocol during ECMO, which included 5000 units of heparin intravenously at the time of ECMO initiation and ongoing, received an LA protocol or no systemic anticoagulation. In addition, the studies should have reported at least one outcome of interest, including hospital mortality, successfully weaned off ECMO, blood transfusions, oxygenator exchange, and thrombotic and hemorrhagic complications. Case reports, reviews, and animal experiments, as well as data that cannot be converted and extracted, were excluded.

## Data extraction

Two independent review authors screened the search results according to the inclusion criteria and extracted relevant data. Conflicts were settled through discussion or, if required, a third review author. The following outcomes were extracted from each paper: author and year of publication, country, study design, sample size and participant characteristics, hospital mortality, successfully weaned off ECMO, and thrombotic and hemorrhagic complications.

## Statistical analysis

For the selected studies, information on all available variables was extracted and entered into a Microsoft Excel database, and meta-analyses were performed using Review Manager 5.3 and R 4.0.2. Odds ratios (ORs) were for the dichotomous outcomes, followed with 95% confidence intervals (CIs). The chi-squared test and $I^2$ statistic were used to assess heterogeneity among the studies in each analysis. In case of the presence of statistical heterogeneity ($P < 0.1$, $I^2 >$ 50%), the random effect model was adopted, while the fixed effect model was used for the absence of statistically significant heterogeneity. Publication bias was supported quantitatively by Funnel plots test and Begg's tests. The meta-analysis was registered at http://www.crd.york. ac.uk/PROSPERO/ (CRD42020202168).

## Results

### Results of study selection

The details of the research identification process are shown in Fig 1. Before July 2020, a total of 153 records were screened from the online databases mentioned previously. A manual search of the reference lists identified 19 additional relevant studies. After exclusion of duplicates, a total of 172 studies remained. Of these, 9 publications were assessed for eligibility. In the end, the data of 553 participants involved in 7 trials reporting the safety and efficacy of LA versus SA were included in the quantitative analysis [14–16, 19, 21–23].

The major characteristics and the results of the quality assessment of the included studies are presented in Table 1. The studies were conducted in 4 different countries between 2015

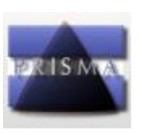

## PRISMA 2009 Flow Diagram

**Identification**

Records identified through database searching
(n= 153)

Additional records identified through other sources
(n = 19)

Records after duplicates removed
(n =172)

**Screening**

Title and abstract excluded (n =156)

Records screened
(n =16)

Records excluded
No anticoagulant targets
Case report (n =7)

**Eligibility**

Full-text articles assessed for eligibility
(n = 9)

Full-text articles excluded with reasons
No grouping information
(n =2)

**Included**

Studies included in qualitative synthesis
(n = 7)

Studies included in quantitative synthesis
(meta-analysis)
(n = 7)

*From:* Moher D, Liberati A, Tetzlaff J, Altman DG, The PRISMA Group (2009). *Preferred Reporting Items for Systematic Reviews and Meta-Analyses: The PRISMA Statement. PLoS Med 6(6): e1000097. doi:10.1371/journal.pmed1000097

**For more information, visit www.prisma-statement.org.**

**Fig 1. Flow diagram for meta-analysis of the comparison of low-dose anticoagulation with standardized dose anticoagulation in patients supported with ECMO.** ECMO = extracorporeal membrane oxygenation.

and 2019. All articles were retrospective observational studies, five were original articles, one was a poster, and one was a letter. In total, data from 553 patients were recorded, of which 255 underwent LA and 298 underwent SA.

## Quality assessment of the included studies

The quality of the included studies was depicted in Fig 2. Attrition bias and reporting bias were rejected in all the included trials. Random sequence generation was depicted clearly in two included trials. The remaining four trials were retrospective controlled studies, and they were unable to describe the generation of random sequences. And allocation concealment was depicted clearly in two included trials. Performance bias and detection bias were hardly avoided since blinding of participants and personnel were hard to conduct and the outcomes were impossible to be assessed blindly.

## Hemorrhagic complications

Meta-analysis demonstrated that the incidence of gastrointestinal tract hemorrhage (Fig 3A) and surgical site hemorrhage (Fig 3B) were both lower in the Low-heparin group compared with the Full-heparin group (OR 0.36, 95% CI 0.20–0.64; OR 0.43, 95% CI 0.20–0.94 respectively), without significant heterogeneity ($I^2 = 0\%$, P = 0.58; $I^2 = 2\%$, P = 0.39, respectively). Whereas there was no significant difference in other hemorrhagic complications, including intracranial hemorrhage (OR 0.62, 95% CI 0.22–1.74) (Fig 3C), pulmonary hemorrhage (OR 0.77, 95% CI 0.30–1.93) (Fig 3D) and ECMO cannula site bleeding (OR 0.38, 95% CI 0.12–1.19) (Fig 3E), without significant heterogeneity in intracranial hemorrhage and pulmonary hemorrhage among the studies ($I^2 = 0\%$, P = 0.94; $I^2 = 0\%$, P = 0.97, respectively). Heterogeneity was observed in ECMO cannula site bleeding among the studies ($I^2 = 60\%$, P = 0.08).

## Thrombotic complications

The present meta-analysis indicated that deep vein thrombosis was similar between the Low-heparin group and the Full-heparin group (OR 0.47, 95% CI 0.16–1.40), without significant heterogeneity ($I^2 = 0\%$, P = 0.85) (Fig 4A). Pulmonary embolism (Fig 4B), clots in the oxygenator and pump (Fig 4C), and intracardiac thrombus (Fig 4D) were all similar between the Low-heparin group and the Full-heparin group (OR 0.79, 95% CI 0.24–2.65; OR 0.67, 95% CI 0.29–

**Table 1. Characteristics of the included studies.**

| Studies | Country | Study types | Number of Patient | | Anticoagulant target | |
|---|---|---|---|---|---|---|
| | | | Low-Heparin | Full-Heparin | Low-Heparin | Full-Heparin |
| Hye Ju 2015 | KR | RCS | 40 | 31 | ACT140–160s | ACT180–220s |
| Zoe 2016 | AUS | RCT | 16 | 15 | ACT140–160s | ACT180–220s |
| Jai Raman 2019 | USA | RCS | 52 | 50 | — | ACT180–220s |
| Katherine L 2019 | USA | RCS | 72 | 131 | — | ACT180–220s |
| Kristen T 2019 | USA | RCS | 23 | 17 | ACT140–160s | ACT180–200s |
| Cécile 2019 | AUS and NZ | RCT | 16 | 16 | APTT < 45s | APTT 50-70s |
| Chitaru 2020 | USA | RCS | 36 | 38 | — | APTT 50-70s |

RCS: retrospective control study; RCT:randomised controlled trial; ACT: activated clotting time of whole blood; APTT: activated partial thromboplastin time.

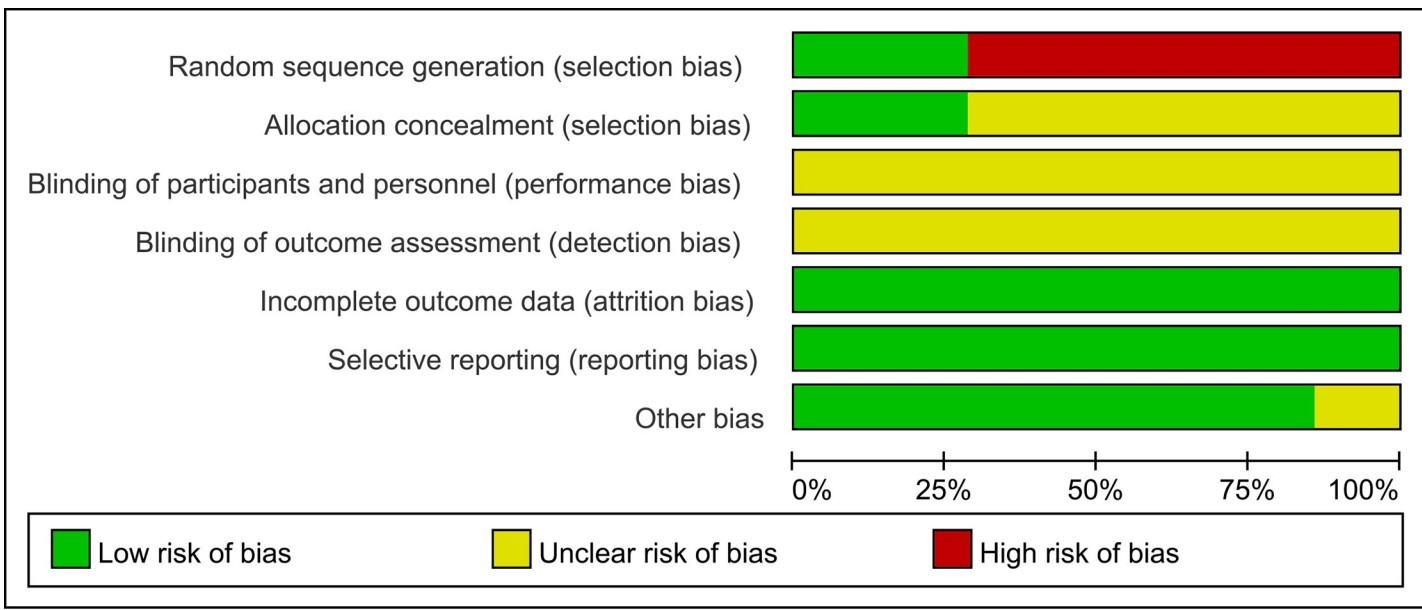

Risk of bias graph: review authors' judgements about each risk of bias item presented as percentages across all included studies.

Risk of bias summary: review authors' judgements about each risk of bias item for each included study.

**Fig 2. Quality assessment of the included studies.**

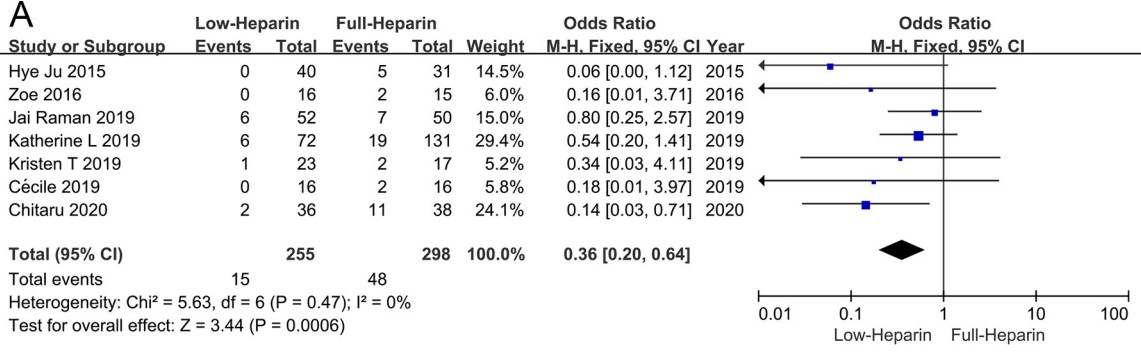

**A**

| Study or Subgroup | Low-Heparin Events | Total | Full-Heparin Events | Total | Weight | Odds Ratio M-H, Fixed, 95% CI | Year |
|---|---|---|---|---|---|---|---|
| Hye Ju 2015 | 0 | 40 | 5 | 31 | 14.5% | 0.06 [0.00, 1.12] | 2015 |
| Zoe 2016 | 0 | 16 | 2 | 15 | 6.0% | 0.16 [0.01, 3.71] | 2016 |
| Jai Raman 2019 | 6 | 52 | 7 | 50 | 15.0% | 0.80 [0.25, 2.57] | 2019 |
| Katherine L 2019 | 6 | 72 | 19 | 131 | 29.4% | 0.54 [0.20, 1.41] | 2019 |
| Kristen T 2019 | 1 | 23 | 2 | 17 | 5.2% | 0.34 [0.03, 4.11] | 2019 |
| Cécile 2019 | 0 | 16 | 2 | 16 | 5.8% | 0.18 [0.01, 3.97] | 2019 |
| Chitaru 2020 | 2 | 36 | 11 | 38 | 24.1% | 0.14 [0.03, 0.71] | 2020 |
| **Total (95% CI)** | | **255** | | **298** | **100.0%** | **0.36 [0.20, 0.64]** | |
| Total events | 15 | | 48 | | | | |

Heterogeneity: Chi² = 5.63, df = 6 (P = 0.47); I² = 0%
Test for overall effect: Z = 3.44 (P = 0.0006)

**B**

| Study or Subgroup | Low-Heparin Events | Total | Full-Heparin Events | Total | Weight | Odds Ratio M-H, Fixed, 95% CI | Year |
|---|---|---|---|---|---|---|---|
| Hye Ju 2015 | 3 | 40 | 9 | 31 | 45.1% | 0.20 [0.05, 0.81] | 2015 |
| Kristen T 2019 | 1 | 23 | 0 | 17 | 2.6% | 2.33 [0.09, 60.84] | 2019 |
| Jai Raman 2019 | 0 | 52 | 2 | 50 | 12.1% | 0.18 [0.01, 3.95] | 2019 |
| Katherine L 2019 | 3 | 72 | 11 | 131 | 36.0% | 0.47 [0.13, 1.76] | 2019 |
| Cécile 2019 | 2 | 16 | 1 | 16 | 4.2% | 2.14 [0.17, 26.33] | 2019 |
| **Total (95% CI)** | | **203** | | **245** | **100.0%** | **0.43 [0.20, 0.94]** | |
| Total events | 9 | | 23 | | | | |

Heterogeneity: Chi² = 4.08, df = 4 (P = 0.39); I² = 2%
Test for overall effect: Z = 2.11 (P = 0.03)

**C**

| Study or Subgroup | Low-Heparin Events | Total | Full-Heparin Events | Total | Weight | Odds Ratio M-H, Fixed, 95% CI | Year |
|---|---|---|---|---|---|---|---|
| Hye Ju 2015 | 1 | 40 | 1 | 31 | 11.5% | 0.77 [0.05, 12.81] | 2015 |
| Zoe 2016 | 0 | 16 | 1 | 15 | 15.7% | 0.29 [0.01, 7.76] | 2016 |
| Cécile 2019 | 0 | 16 | 1 | 16 | 15.2% | 0.31 [0.01, 8.28] | 2019 |
| Kristen T 2019 | 1 | 23 | 0 | 17 | 5.6% | 2.33 [0.09, 60.84] | 2019 |
| Jai Raman 2019 | 0 | 52 | 1 | 50 | 15.9% | 0.31 [0.01, 7.90] | 2019 |
| Katherine L 2019 | 2 | 72 | 5 | 131 | 36.1% | 0.72 [0.14, 3.81] | 2019 |
| **Total (95% CI)** | | **219** | | **260** | **100.0%** | **0.62 [0.22, 1.74]** | |
| Total events | 4 | | 9 | | | | |

Heterogeneity: Chi² = 1.23, df = 5 (P = 0.94); I² = 0%
Test for overall effect: Z = 0.90 (P = 0.37)

**D**

| Study or Subgroup | Low-Heparin Events | Total | Full-Heparin Events | Total | Weight | Odds Ratio M-H, Fixed, 95% CI | Year |
|---|---|---|---|---|---|---|---|
| Hye Ju 2015 | 1 | 40 | 1 | 31 | 10.5% | 0.77 [0.05, 12.81] | 2015 |
| Zoe 2016 | 1 | 16 | 1 | 15 | 9.2% | 0.93 [0.05, 16.39] | 2016 |
| Jai Raman 2019 | 0 | 52 | 1 | 50 | 14.4% | 0.31 [0.01, 7.90] | 2019 |
| Cécile 2019 | 1 | 16 | 1 | 16 | 8.9% | 1.00 [0.06, 17.51] | 2019 |
| Kristen T 2019 | 1 | 23 | 0 | 17 | 5.1% | 2.33 [0.09, 60.84] | 2019 |
| Katherine L 2019 | 3 | 72 | 8 | 131 | 51.8% | 0.67 [0.17, 2.60] | 2019 |
| **Total (95% CI)** | | **219** | | **260** | **100.0%** | **0.77 [0.30, 1.93]** | |
| Total events | 7 | | 12 | | | | |

Heterogeneity: Chi² = 0.83, df = 5 (P = 0.97); I² = 0%
Test for overall effect: Z = 0.56 (P = 0.57)

**E**

| Study or Subgroup | Low-Heparin Events | Total | Full-Heparin Events | Total | Weight | Odds Ratio M-H, Random, 95% CI | Year |
|---|---|---|---|---|---|---|---|
| Hye Ju 2015 | 3 | 40 | 12 | 31 | 30.6% | 0.13 [0.03, 0.51] | 2015 |
| Jai Raman 2019 | 11 | 52 | 21 | 50 | 42.0% | 0.37 [0.16, 0.89] | 2019 |
| Cécile 2019 | 5 | 16 | 4 | 16 | 27.4% | 1.36 [0.29, 6.42] | 2019 |
| **Total (95% CI)** | | **108** | | **97** | **100.0%** | **0.38 [0.12, 1.19]** | |
| Total events | 19 | | 37 | | | | |

Heterogeneity: Tau² = 0.60; Chi² = 4.99, df = 2 (P = 0.08); I² = 60%
Test for overall effect: Z = 1.66 (P = 0.10)

**Fig 3. Forest plots of meta-analysis in hemorrhagic outcomes.** (A) Forest plot of OR of gastrointestinal tract hemorrhage. (B) Forest plot of OR of surgical site hemorrhage. (C) Forest plot of OR of intracranial hemorrhage. (D) Forest plot of OR of pulmonary hemorrhage. (E) Forest plot of OR of ECMO cannula site bleeding. ECMO = extracorporeal membrane oxygenation.

1.58; OR 0.34, 95% CI 0.09–1.30, respectively), all without significant heterogeneity ($I^2$ = 0%, P = 0.85; $I^2$ = 0%, P = 0.70; $I^2$ = 0%, P = 0.91; $I^2$ = 0%, P = 0.91, respectively).

## Hospital mortality

There were 5 articles comparing the hospital mortality between the two groups, including 448 patients (203 patients in the Low-heparin group and 245 patients in the Full-heparin group, respectively). Meta-analysis demonstrated that the hospital mortality was similar between the two groups (OR 0.81, 95% CI 0.42–1.56), and heterogeneity was observed among the studies ($I^2$ = 56%, P = 0.06) (Fig 5A).

## Successfully weaned off ECMO

There were 4 articles comparing the successfully weaned off ECMO rates between the two groups, including 416 patients (187 patients in the Low-heparin group and 229 patients in the Full-heparin group, respectively). Meta-analysis demonstrated that the successfully weaned off ECMO rate was similar between the two groups (OR 0.80, 95% CI 0.30–2.14), and heterogeneity was observed among the studies ($I^2$ = 78%, P = 0.004) (Fig 5B).

## Oxygenator exchange

There were 5 articles comparing the oxygenator exchange between the two groups, including 319 patients (167 patients in the Low-heparin group and 152 patients in the Full-heparin group, respectively). Meta-analysis demonstrated that the oxygenator exchange was similar between the two groups (OR 0.65, 95% CI 0.34–1.24), without significant heterogeneity ($I^2$ = 24%, P = 0.26) (Fig 5C).

## Publication bias analysis

Significant risk of publication bias was not detected of gastrointestinal tract hemorrhage (Fig 6A) and surgical site hemorrhage (Fig 6B), as demonstrated by funnel plots. Begg's tests confirmed there was no significant publication bias (All Pr > |z| >0.05).

## Discussion

As the critical cardiopulmonary support approach to save the lives of critically ill patients originally proposed in the 1970s [24], ECMO has saved thousands of adult and pediatric patients. Nevertheless, patients who underwent ECMO may provoke an inflammatory response in patients who contributes to the contact of blood with the nonendothelial surfaced circuit, leading to consumption and activation of procoagulant and anticoagulant components [25]. Consequently, anticoagulation has traditionally been used to prevent thrombosis of the ECMO circuit; however, it has also increased the risk of excessive bleeding [26]. In addition, large volumes of blood products are used in an effort to decrease bleeding in the setting of anticoagulation, and this leads to significant increases in transfusion-related complications. Supported by advancements in the ECMO system, many centers have gradually attempted the LA strategy, which has achieved gratifying achievements [14–16, 21, 22]. However, whether LA is feasible or superior to SA is still controversial. Thus, we first used the meta-analysis to include

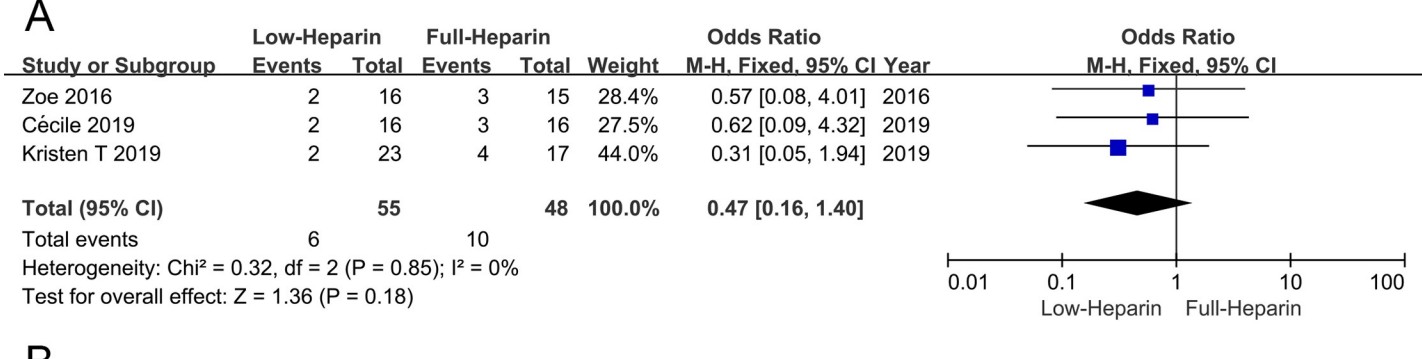

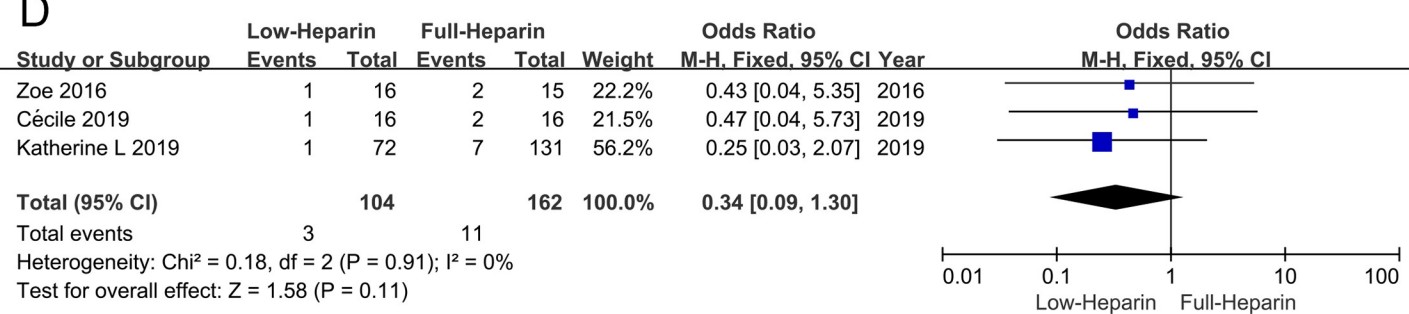

**Fig 4. Forest plots of meta-analysis in thrombotic outcomes.** (A) Forest plot of OR of deep vein thrombosis. (B) Forest plot of OR of pulmonary embolism. (C) Forest plot of OR of clots in the oxygenator and pump. (D) Forest plot of OR of intracardiac thrombus.

comparative studies of the safety and efficacy of LA and SA for patients supported with ECMO and to provide a rational basis for future studies.

As mentioned above, although not all the included studies were randomized controlled trials, most of them were of moderate to high quality. Our study included 7 publications with 553

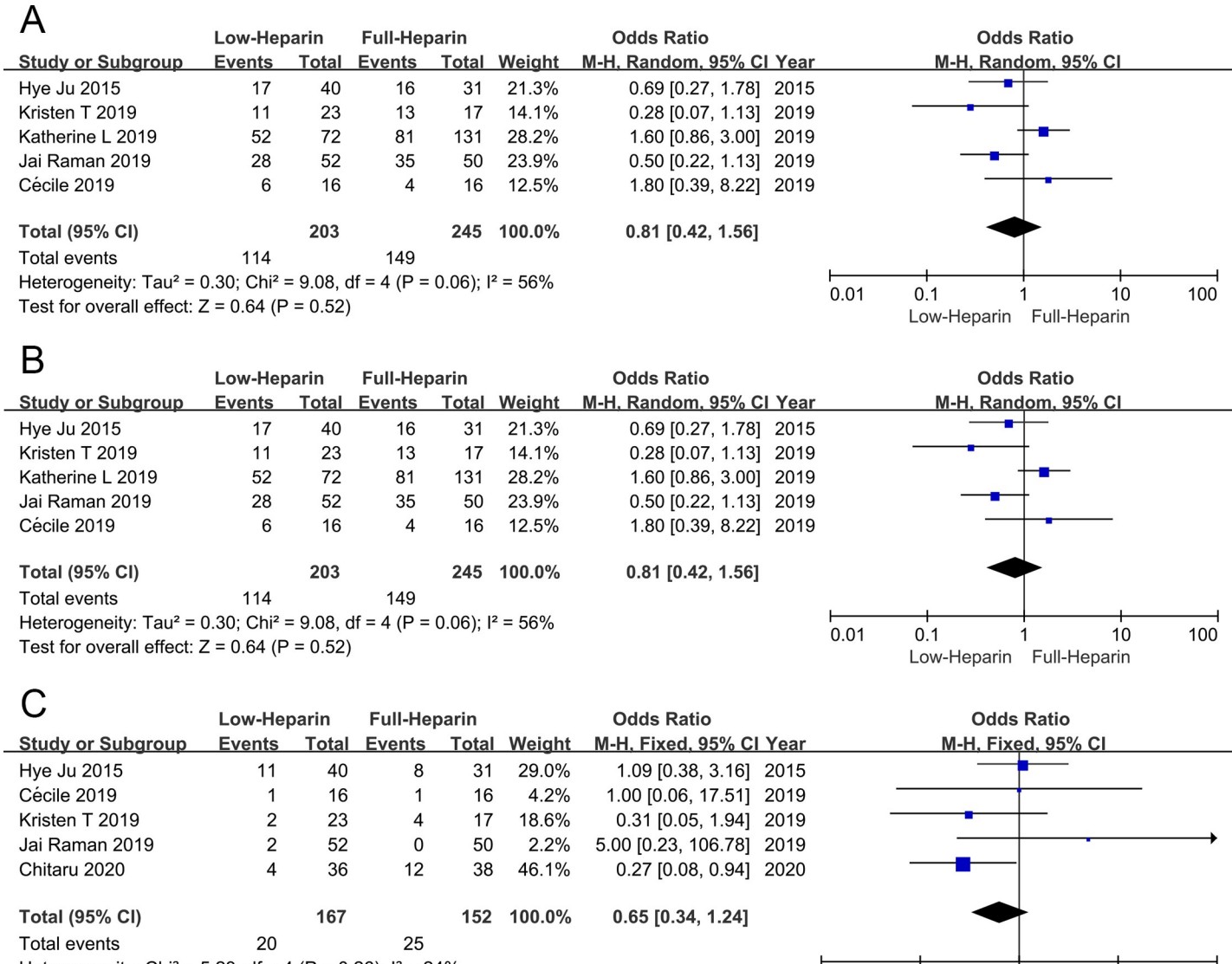

**Fig 5. Forest plots of meta-analysis in hospital mortality (A), successfully weaned off ECMO (B) and oxygenator exchange (C).** ECMO = extracorporeal membrane oxygenation.

patients and reflects the latest results, and we focused on both hemorrhagic and thrombotic complications outcomes of LA and SA for patients receiving ECMO. Through this systematic and comparative evaluation of the safety and efficacy of LA and SA during ECMO, the following conclusions were drawn: (1) LA is superior to SA with regard to hemorrhagic complications; (2) there was no evidence of significant thrombosis complications when LA was compared with SA; and (3) maintenance with LA is safe in patients treated with ECMO.

Anticoagulation-related hemorrhagic complications can be due to excessive anticoagulation. In terms of these complications, including surgical site bleeding and pulmonary, intracranial or gastrointestinal hemorrhage, are inextricably linked to a higher risk of mortality and other complications (e.g., infection, transfusion-related complications, and multiple organ failure) [25, 27, 28]. Notably, a growing number of ECMO centers who have had to run patients

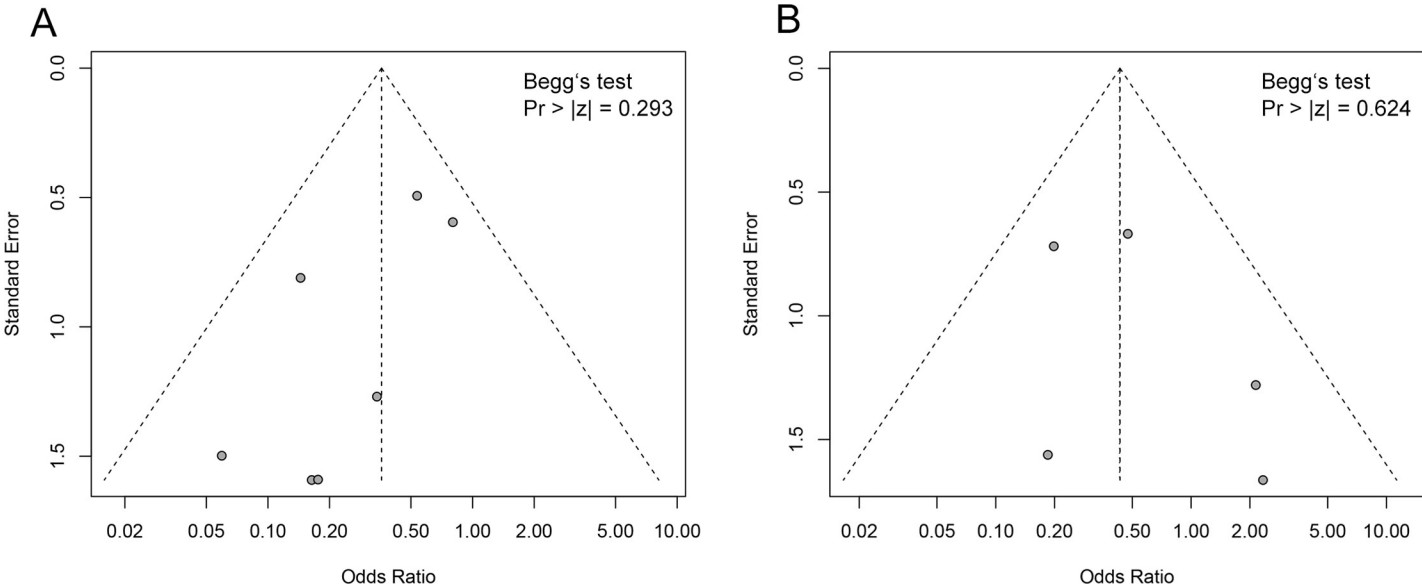

**Fig 6. Funnel plots of gastrointestinal tract hemorrhage (A) and surgical site hemorrhage (B).**

with low doses or no heparin for a variety of reasons (e.g., recent cardiopulmonary bypass, trauma, and head and other organ injury associated with bleeding) [29–32]. Moreover, ECMO with LA or without heparin has been reported in lung transplantation [33, 34]. To the best of our knowledge, this comprehensive review using data from the latest studies to conduct meta-analysis of gastrointestinal tract hemorrhage (OR 0.36, 95% CI 0.20–0.64), surgical site hemorrhage (OR 0.43, 95% CI 0.20–0.94), intracranial hemorrhage (OR 0.62, 95% CI 0.22–1.74), pulmonary hemorrhage (OR 0.77, 95% CI 0.30–1.93) and ECMO cannula site bleeding (OR 0.38, 95% CI 0.12–1.19) for LA and SA in patients supported with ECMO. Alternatively, we cannot compare the volume of blood transfusions between LA and SA to reflect the risk of bleeding, because the blood transfusion volume was not expressed in (mean ± sd). Overall, we can explain the fact that LA seemed to be advantageous over SA with regard to hemorrhagic complications, which confirmed the findings of previous authors [14–16]. The main reason is that gastrointestinal tract bleeding from stress ulceration is extremely common in critically ill patients with coagulopathy [35], especially in patients supported on ECMO. Although this advantage was only limited to the gastrointestinal tract and surgical site hemorrhage in this study, this will bring great benefits to the prognosis and cost, especially with an emphasis placed on increasing the quality of patient care at a decreased cost [25].

Notably, the main concern is that insufficient anticoagulation may lead to thrombosis. However, with the progress of science and technology and the continuous expansion of the scope of application of ECMO, such as some patients with trauma, heparin-induced thrombocytopenia (HIT) and bleeding tendency, the concept of LA has been constantly produced, and at the same time, there was no increase in thrombotic events [33, 36]. A series of case reports have reported encouraging results in LA [10–13, 30–32]. Subsequently, Wood KL et al. reported that the absence of routine systemic anticoagulation for patients supported by VA ECMO was not associated with pump failure, oxygenator failure or thrombotic complications [16]. In addition, Carter KT et al. demonstrated that thrombotic complications did not differ between heparin-sparing and full therapeutic heparin strategies for management of venovenous ECMO [15]. We also indicated the same thrombotic complications rate between the two groups, which indirectly implied the use of LA for ECMO was safe and feasible. Indeed,

microembolic events were invisible to our naked eyes. Marinoni and colleagues have detected microembolic signals by transcranial Doppler in patients treated with ECMO; independently from their pathophysiology, microembolic signals do not seem to influence clinical outcomes [17]. Even so, we cannot rule out the possibility that the lack of anticoagulation may increase subclinical thrombotic events.

Meanwhile, we observed the similarity of hospital mortality and successfully weaned off ECMO rate between the Low-heparin group and the Full-heparin group. Different doses of anticoagulation strategies did not affect the outcome of death. Hospital mortality or successfully weaned off ECMO seems to be more closely related to the severity of the primary disease or the initial physiological state of the patients before ECMO [37, 38]. Fux T et al. believed that a nonshockable rhythm, arterial lactate, and ischemic heart disease were identified as independent pre-VA ECMO risk factors for 90-day mortality [39]. HIT during ECMO can be a significant, life-threatening complication that requires additional resource utilization and has long-term detrimental effects. A multicenter study showed that prevalence of HIT among patients under VA ECMO was extremely low at 0.36%, with an associated mortality rate of 33.3% [40]. Recently, a meta-analysis reported that of 309 patients from six retrospective studies undergoing extracorporeal life support, 83% were suspected of and 17% were confirmed to have HIT [41]. However, only one of the reports included in this study analyzed the results of HIT. Consequently, we could not address the correspondence on this complication in this meta-analysis. More studies are required to evaluate the outcomes of HIT between LA and SA.

## Study limitations

This meta-analysis has several limitations. First, the included studies were all retrospective studies, and only one of them was a randomized controlled trial. The total number of patients was still small, with a greater risk of potential bias. Second, differences exist in individual patient comorbidities, ECMO circuit components and flow hemodynamics. Third, the data we used are based on the published literature, rather than primary data, as we were unable obtain unpublished data.

## Conclusions

Despite the limitations noted, the data confirm that LA in patients treated with ECMO is associated with benefits in hemorrhagic complications and equates in thrombotic complications compared with SA. In particular, LA was found to have significantly lower rates of gastrointestinal tract hemorrhage and surgical site hemorrhage. Meanwhile, the LA strategy for patients supported by ECMO is not associated with thrombotic complications, hospital mortality, or successfully weaned off ECMO rate. These findings seem to support the use of LA for the patients treated with ECMO. Moreover, the LA strategy is advantageous over the SA strategy. Furthermore, larger patient populations in this area are needed to evaluate the safety and efficacy of LA compared with SA in patients receiving ECMO.

## Supporting information

**S1 Checklist. PRISMA 2009 checklist.**
(DOC)

## Author Contributions

**Conceptualization:** Xiaochai Lv.

**Data curation:** Xiaochai Lv, Manjun Deng, Lei Wang, Xiaofu Dai.

**Formal analysis:** Xiaochai Lv, Manjun Deng.

**Funding acquisition:** Liangwan Chen.

**Investigation:** Xiaochai Lv.

**Methodology:** Xiaochai Lv.

**Project administration:** Xiaochai Lv.

**Resources:** Xiaochai Lv.

**Software:** Xiaochai Lv, Manjun Deng.

**Supervision:** Liangwan Chen.

**Validation:** Xiaochai Lv, Lei Wang, Yi Dong, Xiaofu Dai.

**Visualization:** Xiaochai Lv.

**Writing – original draft:** Xiaochai Lv.

**Writing – review & editing:** Xiaochai Lv.

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
