## [Decision Letter · Decision Letter 0]

11 Mar 2021

PONE-D-20-35042

Low vs standardized dose anticoagulation regimens for extracorporeal membrane oxygenation: A Meta-analysis

PLOS ONE

Dear Dr. chen,

Thank you for submitting your manuscript to PLOS ONE. After careful consideration, we feel that it has merit but does not fully meet PLOS ONE’s publication criteria as it currently stands. Therefore, we invite you to submit a revised version of the manuscript that addresses the points raised during the review process.

This is an interesting papers. HOwever, on a methodologic point of view, abstracts and letters should be not considered in the analysis. Please modify. 

We look forward to receiving your revised manuscript.

Kind regards,

Chiara Lazzeri

Academic Editor

PLOS ONE

Journal Requirements:

Reviewers' comments:

Reviewer's Responses to Questions

**Comments to the Author**

1. Is the manuscript technically sound, and do the data support the conclusions?

Reviewer #1: No

2. Has the statistical analysis been performed appropriately and rigorously? 

Reviewer #1: No

3. Have the authors made all data underlying the findings in their manuscript fully available?

Reviewer #1: No

4. Is the manuscript presented in an intelligible fashion and written in standard English?

Reviewer #1: Yes

5. Review Comments to the Author

Reviewer #1: I have two main concerns:

1. Authors present clear criteria for standard doses of heparin (APTT 50-70 sec, ACT 180-220 sec). However there are no criteria for low-dose heparin strategy. What level of APTT, or/and ACT reflect low-dose strategy? If there are no clear criteria, how can authors do any conclusions in comparison of standard vs low doses of heparin during ECMO?

2. For analysis authors use posters and even letters. I’m not sure, that these forms of publications are worthy for meta-analysis.

6. PLOS authors have the option to publish the peer review history of their article (what does this mean?). If published, this will include your full peer review and any attached files.

Reviewer #1: **Yes: **Konstantin A. Popugaev

---

## [Author Response · Author response to Decision Letter 0]

21 Mar 2021

1. Is the manuscript technically sound, and do the data support the conclusions?

Reviewer #1: No

The author replied：I would like to thank the reviewers for their valuable advice. The reviewer replied to no because we included letter in the study. First of all, I would like to thank the reviewers for their review of our manuscript. Meta analysis, as a secondary literature study, may be comprehensive, and whether it can be included depends on the quality of the included study. The letter included in this study meets the inclusion criteria, and it reports the grouping methods, research results and so on. We conducted a bias risk analysis and concluded that it met our inclusion criteria. Please fully consider our point of view, if there are still problems, we can change and re-analyze.

2. Has the statistical analysis been performed appropriately and rigorously?

Reviewer #1: No

The author replied：I would like to thank the reviewers for their valuable advice.The reviewer replied to no because we included letter in the study.

First of all, I would like to thank the reviewers for their review of our manuscript. Meta analysis, as a secondary literature study, may be comprehensive, and whether it can be included depends on the quality of the included study. The letter included in this study meets the inclusion criteria, and it reports the grouping methods, research results and so on. We conducted a bias risk analysis and concluded that it met our inclusion criteria. Please fully consider our point of view, if there are still problems, we can change and re-analyze.

3. Have the authors made all data underlying the findings in their manuscript fully available?

Reviewer #1: No

The author replied：First of all, I would like to thank the reviewers for their review of our manuscript. Meta analysis as a secondary literature study, we have provided all the data that can support our conclusions, and our research is based on published literature. I wonder if there are any other requests. We look forward to hearing from you very much.

4. Is the manuscript presented in an intelligible fashion and written in standard English?

Reviewer #1: Yes

The author replied：Thanks

5. Review Comments to the Author

Reviewer #1: I have two main concerns:

1. Authors present clear criteria for standard doses of heparin (APTT 50-70 sec, ACT 180-220 sec). However there are no criteria for low-dose heparin strategy. What level of APTT, or/and ACT reflect low-dose strategy? If there are no clear criteria, how can authors do any conclusions in comparison of standard vs low doses of heparin during ECMO?

2. For analysis authors use posters and even letters. I’m not sure, that these forms of publications are worthy for meta-analysis.

The author replied to # 1：You have a very good point. In this meta analysis，patients were included in the standard doses of heparin group if they were started on a continuous infusion of anticoagulant after initiation of ECMO, which was monitored with ACT with a target of 180–220 s or APTT target of 50–70 s. The low-heparin group was supported with an low-dose strategy during ECMO, which included 5000 units of heparin intravenously at the time of ECMO initiation and ongoing, received an low-dose protocol (ACT goal: 140–160 s or APTT < 45 s), or did not receive systemic anticoagulation. Since heparin was not used continuously in the follow-up, the bleeding measurements such as ACT or APTT were not continuous monitored. However, compared with the standard dose, it also belongs to low-dose anticoagulation. The purpose of this study was to compare the advantages and disadvantages of low and standardized dose anticoagulation regimens, so it was also included in this study. Please fully consider our point of view.

The author replied to # 2：First of all, I would like to thank the reviewers for their review of our manuscript. Meta analysis, as a secondary literature study, may be comprehensive, and whether it can be included depends on the quality of the included study. The letter included in this study meets the inclusion criteria, and it reports the grouping methods, research results and so on. We conducted a bias risk analysis and concluded that it met our inclusion criteria. Please fully consider our point of view, if there are still problems, we can change and re-analyze.

---

## [Editor Report · Decision Letter 1]

26 Mar 2021

Low vs standardized dose anticoagulation regimens for extracorporeal membrane oxygenation: A Meta-analysis

PONE-D-20-35042R1

Dear Dr. chen,

We’re pleased to inform you that your manuscript has been judged scientifically suitable for publication and will be formally accepted for publication once it meets all outstanding technical requirements.

Kind regards,

Chiara Lazzeri

Academic Editor

PLOS ONE
---

## [Editor Report · Acceptance letter]

31 Mar 2021

PONE-D-20-35042R1 

Low vs standardized dose anticoagulation regimens for extracorporeal membrane oxygenation: A Meta-analysis 

Dear Dr. Chen:

I'm pleased to inform you that your manuscript has been deemed suitable for publication in PLOS ONE. Congratulations! Your manuscript is now with our production department. 

Kind regards, 

on behalf of

Dr. Chiara Lazzeri 

Academic Editor

PLOS ONE